

# Characteristics of patients discontinuing outpatient services under long-term care insurance and its effect on frailty during COVID-19

Tamaki Hirose[1,2,*], Yohei Sawaya[1,2,*], Takahiro Shiba[2],
Masahiro Ishizaka[1], Ko Onoda[1], Akira Kubo[1] and Tomohiko Urano[2,3]

[1] Department of Physical Therapy, School of Health Sciences, International University of Health and Welfare, Otawara, Tochigi, Japan
[2] Nishinasuno General Home Care Center, Department of Day Rehabilitation, Care Facility for the Elderly "Maronie-en", Nasushiobara, Tochigi, Japan
[3] Department of Geriatric Medicine, School of Medicine, International University of Health and Welfare, Narita, Chiba, Japan
* These authors contributed equally to this work.

## ABSTRACT

**Background:** Among community-dwelling older adults who require long-term care and use outpatient rehabilitation services, we aimed to examine the characteristics of patients who discontinued using outpatient rehabilitation services to prevent exposure to COVID-19 and the effects of this discontinuation on patient frailty.

**Methods:** Participants were 119 older adults (69 males, 50 females; average age 77.3 ± 8.3 years) requiring long-term care who used outpatient rehabilitation services. Our outpatient rehabilitation service involved day care for older adults undergoing rehabilitation including pick-up and drop-off services under the long-term care insurance system. They were divided into two groups: participants who discontinued using outpatient rehabilitation services and participants who continued their use. To find the factors associated with this discontinuation, binomial logistic regression analysis was performed, in which the following independent variables were used: gender, age, height, weight, long-term care level, grip strength, and normal walking speed. Frailty status was investigated in April 2020 and again in September 2020 through interviews and telephone surveys utilizing the Frailty Screening Index.

**Results:** Women and normal walking speed were independent factors associated with the discontinuation of outpatient rehabilitation due to COVID-19. According to the results of the Frailty Screening Index, similar tendencies were exhibited in both April and September. The discontinued group indicated that they were getting less exercise and having stronger feelings of fatigue than the continuing group.

**Conclusions:** The findings are in accordance with other studies indicating that women are more likely to employ prevention measures against COVID-19, as seen among the community-dwelling older adults requiring long-term care who used outpatient rehabilitation services. People with higher levels of physical function

Corresponding author
Tomohiko Urano,
turano@iuhw.ac.jp

were also likely to refrain from using outpatient rehabilitation services. Results further suggested that the discontinued group had more frailty-related factors (i.e., low physical activity and strong exhaustion) than the continuing group.

## INTRODUCTION

Since December 2019, infections caused by the novel coronavirus, COVID-19, have been spreading throughout the world (*Zhu et al., 2020*). The World Health Organization (WHO) declared COVID-19 a pandemic in March 2020, and countries around the world began taking various measures to prevent infection, such as locking down cities (*World Health Organization, 2020*). In Japan, a state of emergency was also declared owing to the COVID-19 pandemic; with such a measure, the population of Tochigi Prefecture— the target area of this study—underwent forced restrictions in their daily life, such as refraining from non-essential trips and outings, closure of schools, and working from home. Social connections have been diminished or fragmented, greatly affecting people.

The possibility has been suggested that young people would experience decreased physical activity due to COVID-19, such as a decrease in the number of their steps (*Tison et al., 2020*; *Sun et al., 2020*; *Ammar et al., 2020*). Among older adults, it has been suggested that there would be a reduction in physical activity and a worsening of frailty status due to activity restrictions required to prevent COVID-19 exposure (*Yamada et al., 2020*). Furthermore, COVID-19 has been reported to affect psychological factors, such as stress, sleep, and anxiety (*Casagrande et al., 2020*; *Forte et al., 2020*; *Xiao et al., 2020*). However, most previously presented studies included healthy older adults without underlying illnesses as participants. It is not known what effects activity restrictions due to COVID-19 have on disabled older adults who need long-term support or care.

Many community-dwelling older adults, who need long-term support or care, commute from their homes to use facilities covered by long-term care insurance. During the COVID-19 epidemic, patients who discontinued their use of facilities covered by long-term care insurance were forced to make large changes in their way of life. Because they differ in this way from healthy older adults, shedding light on the unique effects of the pandemic on those who need long-term care or who commute from their homes to use facilities covered by long-term care insurance is an important issue in Japan due to its rapidly aging society.

The purpose of this research was to study community-dwelling older adults needing long-term support who utilize outpatient rehabilitation services and to shed light on the characteristics of patients who discontinued their use of outpatient rehabilitation services because of COVID-19 prevention efforts. It also aimed to examine differences

regarding frailty level between discontinuation and continuation of outpatient rehabilitation services.

## MATERIALS AND METHODS

### Participants

As of April 1, 2020, the outpatient rehabilitation service, located in Tochigi prefecture, had 143 of its patients registered in this prospective observational study. Our outpatient rehabilitation service involved day care for older adults undergoing rehabilitation including pick-up and drop-off services under the long-term care insurance system. The rehabilitation program includes muscle strengthening exercises (by machine training), exercise therapy (using an ergometer), activities of daily living/movement guidance, among others. To determine eligible participants, the following individuals were excluded: 12 users whose walking speed and grip strength could not be measured, six users for whom data were unavailable, four users who were absent long-term for a reason other than COVID-19, one user who took 141 s for 5-m walking test and one user over the age of 100. The participant group, therefore, included 119 patients needing long-term support and care (69 males, 50 females, aged 77.3 ± 8.3 years: average value ± standard deviation) and who used outpatient rehabilitation services at least once a week (Fig. 1). All participants were provided written explanations concerning the assessments, and written informed consent was obtained. This study obtained approval from the International University of Health and Welfare Institutional Review Board (Approval No.: 17-Io-189-6) and was conducted in accordance with the guidelines proposed by the Declaration of Helsinki.

### Assessments

The participants were divided into two groups: one included those who discontinued use of outpatient rehabilitation services because of COVID-19 prevention ("discontinued group"); the other included those who continued use ("continuing group"). The definition of discontinued was a 2-week or longer absence from outpatient rehabilitation during the period of the nationwide state of emergency declaration in Japan (April 7, 2020–May 25, 2020) for COVID-19 prevention. The basic attributes (age, height, weight, and long-term care level), grip strength, and normal walking speed of all participants were referenced from information in their patient charts dated March and April 2020.

Grip strength was measured using a digital dynamometer (TKK 5401 Grip-D, Takei Scientific Instruments, Niigata, Japan). Measurement was performed with the participants seated in a relaxed position (not resting on the back of the chair), holding the dynamometer to the side of the body in an extended elbow position. Measurements were taken twice on both the left and right sides, with the maximum values of the right and left sides having been used as the representative values.

To measure walking speed, the participants were asked to walk at their normal walking speed along an 11 m straight course with marks on the floor indicating 3 m and 8 m. The time taken to walk between the 3 m and 8 m marks was measured. The selection of the conditions for the assessment of normal walking speed and the distribution of acceleration
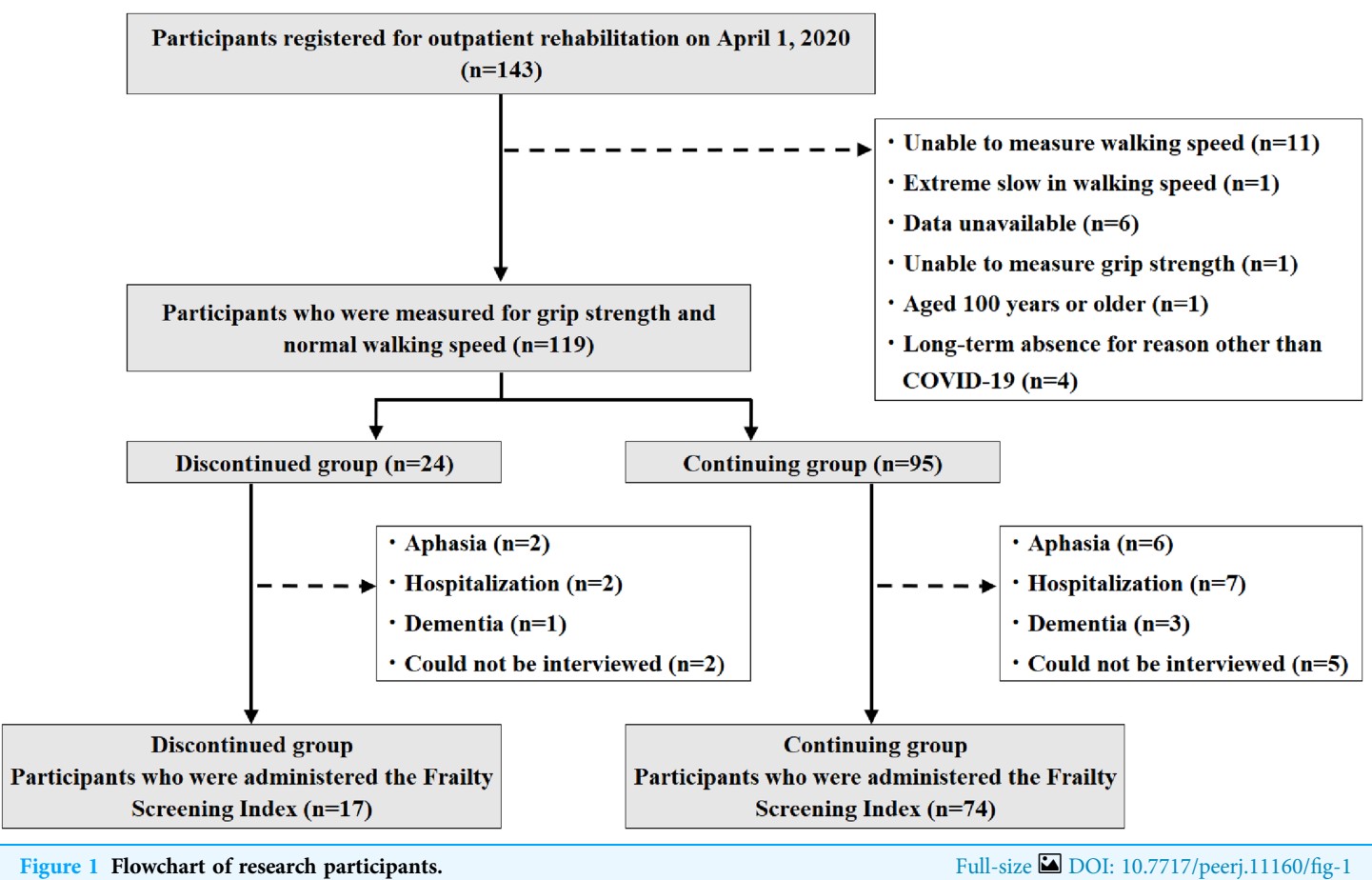

**Figure 1 Flowchart of research participants.**

and deceleration paths were guided by prior established guidelines (*Arai, 2018*; *Satake & Arai, 2020*).

Frailty status was assessed twice, in April 2020 and in September 2020, through questionnaires and telephone surveys utilizing the Frailty Screening Index. Those who had terminated the use of outpatient rehabilitation were interviewed by telephone. Based on prior studies, the Frailty Screening Index asks the following questions: for weight loss, "Have you lost 2 kg or more in the past 6 months?" (Yes = 1 point); for walking speed decline, "Do you think you walk slower than before?" (Yes = 1 point); for exercise, "Do you go for a walk for your health at least once a week?" (No = 1 point); for memory, "Can you recall what happened 5 minutes ago?" (No = 1 point); and for feeling fatigued, "In the last 2 weeks, have you felt tired without a reason?" (Yes = 1 point). A score of three or more was defined as frailty, a score of one or two was defined as pre-frailty, and a score of zero was defined as robust (*Yamada & Arai, 2015*). Before the Frailty Screening Index survey was used, participants that had aphasia, dementia, were hospitalized, or could not be interviewed over the telephone were excluded (Fig. 1).

## Statistical analyses

The statistical analysis methods used for comparing the discontinued group and the continuing group were the unpaired *t*-test for age, height, weight, grip strength, and

**Table 1** Basic attributes of the participants.

| | Discontinued group (n = 24) | Continuing group (n = 95) | p value |
|---|---|---|---|
| Age (years) | 76.8 ± 8.5 | 77.4 ± 8.3 | 0.747 |
| Number by female sex | 12 | 38 | 0.488 |
| Long-term care level | | | |
| Needs support 1 | 8 (33) | 17 (18) | 0.080 |
| Needs support 2 | 5 (21) | 11 (12) | |
| Needs care 1 | 5 (21) | 36 (38) | |
| Needs care 2 | 4 (17) | 20 (21) | |
| Needs care 3 | 2 (8) | 9 (9) | |
| Needs care 4 | 0 | 2 (2) | |
| Needs care 5 | 0 | 0 | |
| Height (cm) | 160.8 ± 7.2 | 159.1 ± 8.9 | 0.380 |
| Weight (kg) | 58.7 ± 10.1 | 58.5 ± 12.2 | 0.940 |
| Grip strength (kg)[†] | 23.2 ± 6.9 | 22.4 ± 8.7 | 0.673 |
| Normal walking speed (m/s) | 0.85 ± 0.4 | 0.59 ± 0.3 | 0.001* |

Notes:
* $p < 0.05$.
[†] Measurements were taken twice on both the left and right sides, with the maximum values of the right and left sides having been used as the representative values.
Average Value ± Standard Deviation. N (%).
Discontinued group: Participants who discontinued use of outpatient rehabilitation for two or more weeks because of COVID-19 prevention.
Continuing group: Participants who continued use of outpatient rehabilitation.

normal walking speed, Fisher's exact test for gender, and the Wilcoxon rank sum test for long-term care level. A seven-stage ordinal scale was used for long-term care level (*Ministry of Health, Labour and Welfare, 2016*). Furthermore, to find the associated factors for the discontinued group, using 0 as a dependent variable for the continuing group and 1 as a dependent variable for the discontinued group, binomial logistic regression analysis was carried out with basic attributes (gender, age, height, weight, and long-term care level), grip strength, and normal walking speed as independent variables. For gender, males were 0 and females 1. In addition, the Frailty Screening Index of the discontinued group and the continuing group were compared in April and again in September. Responses to each question and frailty levels were compared using Fisher's exact test. IBM SPSS Statistics V25.0 was used for all statistical analyses, with a significance level of 5%.

## RESULTS

Figure 1 depicts the flowchart of research participants: 24 in the discontinued group and 95 in the continuing group. The participants attributes are shown in Table 1. There was a significant difference between the normal walking speeds of the discontinued group and the continuing group. Binomial logistic regression analysis was used to find associated factors related to the discontinuation of outpatient rehabilitation due to COVID-19 (Table 2). Compared to the continuing group, the discontinued group had a significantly higher normal walking speed and was an independent variable for discontinuation of use

**Table 2 Factors related to discontinuing use of outpatient rehabilitation to prevent COVID-19 (binomial logistic regression analysis).**

|  | OR | 95% CI | p value |
|---|---|---|---|
| Age (years) | 0.987 | [0.924–1.055] | 0.701 |
| Gender (Female) | 5.125 | [1.135–23.144] | 0.034* |
| Level of care | 0.970 | [0.632–1.489] | 0.890 |
| Height (cm) | 1.104 | [1.000–1.219] | 0.050 |
| Weight (kg) | 0.982 | [0.930–1.037] | 0.509 |
| Grip strength (kg)† | 0.995 | [0.908–1.090] | 0.907 |
| Normal walking speed (m/s) | 11.664 | [2.069–65.752] | 0.005* |

Notes:
* $p < 0.05$.
† Measurements were taken twice on both the left and right sides, with the maximum values of the right and left sides having been used as the representative values.
Dependent variables: Continuing group = 0, Discontinued group = 1.
Independent variables: Male = 0, Female = 1.
OR, Odds ratio; CI, Confidence interval.
Nagelkerke $R^2$ = 0.215.

**Table 3 Comparison of Frailty Screening Index for continuing group and discontinued group by time period.**

| Item | Questions | Answer | | April 2020 frailty screening index | | | September 2020 frailty screening index | | |
|---|---|---|---|---|---|---|---|---|---|
|  |  |  |  | Discontinued group (n = 17) | Continuing group (n = 74) | p value | Discontinued group (n = 17) | Continuing group (n = 74) | p value |
| Weight Loss | Have you lost 2 kg or more in the past 6 months? | No | 0 | 14 (82) | 60 (81) | 1.000 | 13 (76) | 59 (80) | 0.748 |
|  |  | Yes | 1 | 3 (18) | 14 (19) |  | 4 (24) | 15 (20) |  |
| Low physical function | Do you think you walk slower than before? | No | 0 | 11 (65) | 49 (66) | 1.000 | 10 (59) | 47 (64) | 0.784 |
|  |  | Yes | 1 | 6 (35) | 25 (34) |  | 7 (41) | 27 (36) |  |
| Low physical activity | Do you go for a walk for your health at least once a week? | Yes | 0 | 10 (59) | 68 (92) | 0.002* | 14 (82) | 72 (97) | 0.043* |
|  |  | No | 1 | 7 (41) | 6 (8) |  | 3 (18) | 2 (3) |  |
| Cognition | Can you recall what happened 5 minutes ago? | Yes | 0 | 16 (94) | 70 (95) | 1.000 | 16 (94) | 69 (93) | 1.000 |
|  |  | No | 1 | 1 (6) | 4 (5) |  | 1 (6) | 5 (7) |  |
| Exhaustion | In the last 2 weeks, have you felt tired without a reason? | No | 0 | 10 (59) | 63 (85) | 0.037* | 10 (59) | 64 (86) | 0.015* |
|  |  | Yes | 1 | 7 (41) | 11 (15) |  | 7 (41) | 10 (14) |  |
| Robust |  |  |  | 5 (29) | 35 (47) | 0.154 | 6 (35) | 32 (43) | 0.145 |
| Pre-frailty |  |  |  | 8 (47) | 33 (45) |  | 8 (47) | 39 (53) |  |
| Frailty |  |  |  | 4 (24) | 6 (8) |  | 3 (18) | 3 (4) |  |

Notes:
N (%).
* $p < 0.05$.

(Table 2: OR = 11.664, 95% CI [2.069–65.752]). Another independent variable for discontinuation of use identified was female gender (OR = 5.125, 95% CI [1.135–23.144]).

Finally, Table 3 shows the Frailty Screening Index analysis results for both the discontinued and continuing groups, conducted in April and September 2020. Looking at each item of the Frailty Screening Index questionnaire administered in April 2020, participants in the discontinued group tended to have more frequent responses of "No" to the question, "Do you go for a walk for health reasons at least once a week?" They were also

more likely to answer "Yes" to the question, "In the last 2 weeks, have you felt tired without a reason?" compared to the continuing group ($P = 0.002$ and $P = 0.037$). A statistically significant difference was also found in the answers to questions about exercise and feeling fatigued in the survey conducted in September ($P = 0.043$ and $P = 0.015$, respectively). Despite identifying two significant questions, we observed no significant differences between participants of the two groups in terms of them being generally frailty, pre-frailty, or robust (as previously defined).

## DISCUSSION

From the results, "female" and "fast walking speed" were extracted as characteristics of patients who discontinued use of outpatient rehabilitation services because of COVID-19 prevention. According to a prior study, females implemented infection countermeasures and prevention against COVID-19 to a greater extent than males; even the oldest amongst the women implemented the countermeasures (*Muto et al., 2020*). That study also reported that females gathered more information than males about COVID-19 (*Muto et al., 2020*). The results of the present study also indicate that females make more efforts to prevent COVID-19 than males. Furthermore, a factor in the voluntary restraint of activities was walking speed (relating to mobility), rather than grip strength (relating to muscular strength). We believe this to be a new finding.

Next, from the Frailty Screening Index analysis results, the group that had discontinued outpatient rehabilitation in April had fewer opportunities for exercise and had stronger feelings of fatigue compared to the continuing group. We believe that the primary cause of the reduction in exercise opportunities was that the participants had discontinued using outpatient rehabilitation services. Additionally, as a factor in feeling fatigued, it has been reported that for people aged 65 or older, exercising with others is linked to preventing depression (*Kanamori et al., 2018*). In other words, it can be inferred that, as opportunities for exercising with others decreased for the discontinued group, their feelings of fatigue would become stronger, and they would tend to become depressed. Moreover, it has been reported that older adults who do not participate in cultural or community activities become frail (*Yoshizawa et al., 2019*). We believe that the role outpatient rehabilitation services play in the lives of older adults requiring long-term support and care is not simply limited to exercise; it also assumes the roles of interpersonal interaction and cultural activity. Furthermore, the group that had discontinued outpatient rehabilitation in September had similarly fewer opportunities for exercise and had stronger feelings of fatigue compared to the continuing group. The nationwide state of emergency declaration was lifted in Japan in May, but COVID-19 shows no signs of remission or resolution; thus, the discontinued group might continue their voluntary restraint of activities for another 6 months or more over the continuing group.

The results of this study have shed light on the characteristics of COVID-19 prevention among older adults who need long-term support or care and utilize outpatient rehabilitation services and the effect of prevention measures on their frailty. A prior study has reported that a comparison of before and after the spread of COVID-19 indicated that weekly physical activity duration was reduced by approximately 30%,

regardless of the presence of frailty (*Zhu et al., 2020*). From the results of this research and prior studies, voluntary restraints on activities due to COVID-19 prevention affects the decline in mental and physical functioning among community-dwelling older adults. Furthermore, our results suggest the need for support and care for older adults because their discontinuation of the use of outpatient rehabilitation services due to COVID-19 prevention may evoke even further deterioration of mental and physical function and will continue to affect frailty. In the future, we believe that the implementation of exercise and social interaction support in a non-contact format for community-dwelling older adults who need long-term support or care would serve as a method for preventing the worsening of frailty status, such as isolation and disuse syndrome, which occur when there is a widespread outbreak of a serious infectious disease.

A limitation of this study was that it was a single center study; thus, the number of participants was small and gender ratio bias may have occurred. Going forward, it would be desirable to explore the long-term impact of discontinuing the use of outpatient rehabilitation services by exploring a larger facility and increasing the number of participants. Furthermore, because this study required participants to undergo telephone-based investigations, we chose to use the Frailty Screening Index, which is a measurement tool for assessing frailty that can be easily administered. Another reason for choosing this index was the report of the Clinical Guide for Frailty of Japan, which described the tool as valid (*Satake & Arai, 2020*).

## CONCLUSIONS

In conclusion, we found that, among the community-dwelling elderly people requiring long-term care who were using outpatient rehabilitation services, women and people with higher levels of physical function were refraining from physical activities. Furthermore, our findings suggested that frailty status in the discontinued group was lower than that in the continuing group.

## ACKNOWLEDGEMENTS

We would like to extend our deepest gratitude to the users and staff members of the Nishinasuno General Home Care Center that cooperated with the implementation of this study.

### Funding

This study was funded by the JSPS Grants-in-Aid for Scientific Research (20K07789 and 20K23204). The funders had no role in study design, data collection and analysis, decision to publish, or preparation of the manuscript.

### Grant Disclosures

The following grant information was disclosed by the authors:
JSPS: 20K07789 and 20K23204.

## Competing Interests

The authors declare that they have no competing interests.

## Author Contributions

- Tamaki Hirose conceived and designed the experiments, performed the experiments, analyzed the data, prepared figures and/or tables, authored or reviewed drafts of the paper, and approved the final draft.
- Yohei Sawaya conceived and designed the experiments, performed the experiments, analyzed the data, prepared figures and/or tables, authored or reviewed drafts of the paper, and approved the final draft.
- Takahiro Shiba conceived and designed the experiments, performed the experiments, authored or reviewed drafts of the paper, and approved the final draft.
- Masahiro Ishizaka conceived and designed the experiments, analyzed the data, authored or reviewed drafts of the paper, and approved the final draft.
- Ko Onoda conceived and designed the experiments, authored or reviewed drafts of the paper, and approved the final draft.
- Akira Kubo conceived and designed the experiments, authored or reviewed drafts of the paper, and approved the final draft.
- Tomohiko Urano conceived and designed the experiments, performed the experiments, analyzed the data, authored or reviewed drafts of the paper, and approved the final draft.

## Human Ethics

The following information was supplied relating to ethical approvals (i.e., approving body and any reference numbers):

This study obtained approval from the International University of Health and Welfare Institutional Review Board (Approval No.: 17-Io-189-6).

## Data Availability

Raw data are available as a Supplemental File.

## Supplemental Information

Supplemental information for this article can be found online at http://dx.doi.org/10.7717/peerj.11160#supplemental-information.

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
