# Peer review of "Characteristics of patients discontinuing outpatient services under long-term care insurance and its effect on frailty during COVID-19"

_PeerJ, doi:10.7717/peerj.11160_

## Round 0.1 · original submission · Major Revisions

You can start your revision as suggested.

Reviewer 1 ·

Basic reporting

no comment

Experimental design

The impact of COVID-19 varies by region. Please describe the impact of COVID-19 around your facility briefly in Introduction section.

Please describe the outpatient rehabilitation program briefly in Materials & Methods section.

Generally older females use outpatient rehabilitation services more frequently than older males. Please describe why the number of older females was less than that of older males in your study in Discussion section.

Validity of the findings

Although Frailty Screening Index is a screening tool for detecting frailty status, it is not the most suitable assessment tool to detect the frailty status change. Authors should describe it as a limitation.

Additional comments

Thank you for the opportunity to review this article. Our health, psychological status and social activity have been significantly affected by COVID-19 pandemic. The study of the effects of discontinuation of outpatient rehabilitation services for older adults during COVID-19 pandemic is an important subject.

Line 157-159
Authors should describe the following sentence in Materials & Methods section.
“Before the Frailty Screening Index survey was used, participants that had aphasia, dementia, were hospitalized, or could not be interviewed over the telephone were excluded (Figure 1)”.

Line 222-225
The following sentence is not necessary in Conclusions section.
“The purpose of this research was to study community-dwelling older adults needing long-term support who utilize outpatient rehabilitation services to shed light on the characteristics of patients who discontinued their use of outpatient rehabilitation services because of COVID-19 prevention efforts.”

Reviewer 2 ·

Basic reporting

The article is clear and well written. The manuscript introduction, background, structure, tables, and figures meet PeerJ standards. The raw data are provided and easily understandable. Indeed, in my opinion, the article needs only some minor revisions to improve its readability (please, see General comments).

Experimental design

This research is within the aim and scope of the journal and the research question is well defined and contextualized within the current literature. The investigation was conducted with the technical and ethical rigor by the journal. Although some minor revisions are necessary, the methods are described properly (please, see General comments).

Validity of the findings

Notwithstanding the relatively small sample size considering the statistical analysis performed to test the primary aim (i.e., binomial logistic regression analysis), I find the primary aim results of this article robust and the statistical analysis sound considering the observational nature of the study. In this regard, I suggest stating the type of study (e.g., observational, prospective, retrospective, etc.) within the manuscript.
As previously mentioned, the raw data are provided and easily understandable.
Overall, the conclusions are well stated, even though I recommend rewording some sentences (please, see General comments).

Additional comments

Line 34. Please, consider changing “Factor analysis”. Although I understand its meaning and it is grammatically correct, “factor analysis” might create misunderstanding due to the statistical analysis called “factor analysis”.

Line 48. Is it refrain from “physical activities” or “outpatient rehabilitation services”?

Line 49-50. Please, consider rephrasing this sentence “Furthermore, the results suggest that frailty status in the discontinued group might worsen compared to the continuing group, with declines in mental and physical function”. Specifically, due to the observational nature of the study, it is difficult to define what is the cause or the effect; hence, terms like “might worsen” and “declines in” might be inappropriate and could be changed in “is worse” and “lower”. Moreover, I do not understand why the authors report declines in mental and physical function? When rephrasing this sentence, please consider Table 3 results.


Line 59-61. Any reference to support this claim?

Line 78. Please, consider rephrasing this sentence “The effect of discontinuation on frailty was examined” (see comment line 49-50).


Line 78-80. Please, consider changing the position of this sentence “and measures were considered to prevent the worsening of frailty status, such as isolation and disuse syndrome”. In the present form, readers might think that the authors have some results regarding the “measures to prevent…“, whereas those measures are only present in this article’s discussion.

Line 98. Please, consider changing the title “Evaluation of frailty” into something that also includes the other concept within the paragraph (e.g., Assessments).

Line 104. How is the “long-term care level” measured? Please, provide the reference if available.

Line 108-109. What was the elbow position (angle) when the grip strength was measured?

Line 110-112. Any reference regarding the methods used to measure the normal walking speed?

Line 114-116. Isn’t this concept “The participants who continued using outpatient rehabilitation were interviewed in September as well” already explain in the previous sentence?

Line 130-131. “to find the factors for the discontinued group” seems incomplete to me. I would expect to find a verb like associated, affect, or predict. Please, consider the term according to the observational nature of the study.

Line 134-136. Please, consider changing the sentence “In addition, the discontinued group and the continuing group were compared against the Frailty Screening Index in April and again in September” into something like “In addition, the Frailty Screening Index of the discontinued group and the continuing group were compared in April and again in September” to clarify that comparisons were performed only between group and not over time.

Line 137. Why the “presence” of frailty? Were the authors referring to frailty levels or scores?

Line 145-146. Please, consider reporting the overall goodness of fit of the binomial logistic regression (e.g., Cox and Snell R square or Nagelkerke R square).

Line 167. Please, consider changing “participants” into something like “participants of the two groups”.

Line 192. Please, consider changing or deleting “From the above results” to avoid giving the impression that the authors are using the results of this study to support the statement that starts with “we believe …”, which, otherwise, I find appropriate within the discussion.

Line 199. Please, consider changing “suggesting” into something less strong and direct.

Line 210. Please, consider changing “will cause” into something less strong and direct.

Line 227-228. Please, consider rephrasing this sentence “Furthermore, ….” (see comment line 49-50).

Figure 1. Please, consider changing “participants measured”. It could be similar or equal to the title that will replace “Evaluation of frailty” (please, see comment line 98).

Table 1. Please, clarify what the grip strength values represent (e.g., average of the right and left arm). If necessary, also edit the method section and Table 2.

Table 3. I have difficulties understanding what statistical test was used to compute the p values in the last row (i.e., 0.087 and 0.281). If Wilcoxon rank sum tests were used to compare the Frailty Screening Index and not the number of participants in each group, I would recommend adding a row with that index’s descriptive information with their p value on the same row.

---

## Round 0.2 · Minor Revisions

Consider this last Reviewer comment: The time of "5m walking test" of No. 247184 was 141.58 in the Supplementry_file1(raw_data).
Please check again if you should include the data of No. 247184 in the analysis or not.

Reviewer 1 ·

Basic reporting

No comment

Experimental design

No comment

Validity of the findings

The time of "5m walking test" of No. 247184 was 141.58 in the Supplementry_file1(raw_data).
Please check again if you should include the data of No. 247184 in the analysis or not.

Additional comments

I confirmed the revisions of your manuscript.
Thank you for your very careful and extensive revision.

Reviewer 2 ·

Basic reporting

No comment.

Experimental design

No comment.

Validity of the findings

No comment.

Additional comments

Thank you for your willingness to address each recommendation and comment I made. I appreciate the time and effort invested in the edits you made. I am satisfied with the authors' responses.

---

## Round 0.3 · accepted · Accept

Thank you for addressing the final comment.